# Cropping outperforms dropout as an augmentation strategy for self-supervised training of text embeddings

**Rita González-Márquez**                    *rita.gonzalez-marquez@uni-tuebingen.de*
*Hertie Institute for AI in Brain Health*
*University of Tübingen, Germany*

**Philipp Berens**                    *philipp.berens@uni-tuebingen.de*
*Hertie Institute for AI in Brain Health*
*University of Tübingen, Germany*

**Dmitry Kobak**                    *dmitry.kobak@uni-tuebingen.de*
*Hertie Institute for AI in Brain Health*
*University of Tübingen, Germany*

**Reviewed on OpenReview:** *https://openreview.net/forum?id=gVRsIh9x7W*

## Abstract

Text embeddings, i.e. vector representations of entire texts, play an important role in many NLP applications, such as retrieval-augmented generation, clustering, or visualizing collections of texts for data exploration. Currently, top-performing embedding models are derived from pre-trained language models via supervised contrastive fine-tuning. This fine-tuning strategy relies on an external notion of similarity and annotated data for generation of positive pairs. Here we study *self-supervised* fine-tuning and systematically compare the two most well-known augmentation strategies used for fine-tuning text embeddings models. We assess embedding quality on MTEB and additional in-domain evaluations and show that cropping augmentation strongly outperforms the dropout-based approach. We find that on out-of-domain data, the quality of resulting embeddings is substantially below the supervised state-of-the-art models, but for in-domain data, self-supervised fine-tuning can produce high-quality text embeddings after very short fine-tuning. Finally, we show that representation quality increases towards the last transformer layers, which undergo the largest change during fine-tuning; and that fine-tuning only those last layers is sufficient to reach similar embedding quality.

## 1 Introduction

Representing texts as vectors is important in natural language processing for both supervised (e.g. spam detection, sentiment analysis, semantic matching) and unsupervised (e.g. clustering, visualization, retrieval) downstream tasks. With the advent of retrieval-augmented generation (RAG), it has become increasingly important to produce high-quality text representations that enable retrieval of relevant texts that can be added to the prompt to improve performance of generative large language models (LLMs) (Izacard et al., 2022b; Ram et al., 2023). Such representations (or text *embeddings*) can be obtained with a wide range of methods, from simple bag-of-words representations such as TF-IDF (Jones, 1972) to transformer-based large language models (see Zhao et al., 2023, for a survey). Such language models are initially trained (*pre-trained*) with a token-level loss, and subsequent fine-tuning with a text-level loss is needed to obtain useful text-level representations (Xu et al., 2023). We refer to models and representations fine-tuned for representing entire texts as *sentence transformers* and *sentence* or *text embeddings*, following Reimers & Gurevych (2019).

In recent benchmarks such as MTEB (Muennighoff et al., 2023), sentence transformers relying on extensive supervised contrastive fine-tuning on large curated datasets have typically performed best, whereas models

solely trained with self-supervised contrastive learning scored worse results. This is in stark contrast to computer vision, where self-supervised learning (SSL) via data augmentations has been immensely successful in producing semantically meaningful image representations (Balestriero et al., 2023). Various SSL approaches have been suggested for sentence-level fine-tuning, including the dropout-based method employed by SimCSE (Gao et al., 2021) and the cropping-based method employed by DeCLUTR (Giorgi et al., 2021). However, it remains unclear how their performance compares between each other and to the state-of-the-art (SOTA) embedding models, especially when evaluated on texts from some specific limited domain.

In this work we address these questions by systematically comparing the two most well-known data-augmentation strategies used for training text embeddings, and by comparing them to the state-of-the-art supervised models. By extensively assessing representation quality on MTEB tasks and additional in-domain evaluations, we show that:

- using text crops as positive pairs for contrastive learning performs consistently better than the dropout-based augmentation used by SimCSE, contrary to the claims in Gao et al. (2021);

- on out-of-domain data, the quality of resulting embeddings is substantially below the supervised SOTA models, but for in-domain data, self-supervised fine-tuning produces high-quality sentence embeddings, comparable to those by the supervised SOTA models;

- self-supervised fine-tuning on a minimal amount of data (as few as 10 000 short input texts) can already lead to large improvements in sentence embedding quality;

- a large part of the improvement during SSL fine-tuning is due to the generic and domain-independent adaptation that can be called *sentence adaptation*;

- representation quality increases towards the last transformer layers, which undergo the largest change during fine-tuning; and fine-tuning only those layers leads to similar embedding quality.

Our findings are noteworthy given that dropout augmentations of SimCSE (Gao et al., 2021) represent one of the most well-known and well-cited SSL approaches in the literature on text embeddings[1]. Dropout augmentations are still being used to fine-tune modern models (e.g. BehnamGhader et al., 2024) and have also been adopted in other fields, e.g. for representation learning of single-cell RNA-sequencing data (Yang et al., 2022).

We therefore believe that, even though modern text-embedding models are usually relying on supervised fine-tuning, systematically comparing different augmentation approaches for self-supervised fine-tuning can provide valuable insights into which training strategies yield the best text representations.

## 2 Related work

### 2.1 Text embeddings from generative and non-generative language models

Early text embeddings were obtained with bag-of-words approaches, such as TF-IDF (Jones, 1972) and BM25 (Robertson et al., 1995; Kamphuis et al., 2020). Transformer-based (Vaswani et al., 2017) language models led to a shift toward dense embeddings, which capture richer semantic information and context compared to the sparse bag-of-words representations. Initially, non-generative encoder-only (i.e. BERT-like) models were used for generating text embeddings since they use bidirectional attention, which is more appropriate for embedding tasks. However, recent work has increasingly adopted generative decoder-only (i.e. GPT-like) models for embedding tasks to leverage powerful recent models with more extensive pre-training.

**Encoder-only models** These models receive a sequence of text tokens as input and produce a separate latent representation for each of the tokens as output, without any restrictions on the attention patterns, resulting in bidirectional self-attention. The BERT model (Devlin et al., 2019) and its later variants such as

---

[1] At the moment of publication, Gao et al. (2021) has 4 900 citations in Google Scholar, second only after Reimers & Gurevych (2019) with 22 300 citations, in the literature on text/sentence embeddings.

RoBERTa (Liu et al., 2019) and MPNet (Song et al., 2020) include an additional classification token `[CLS]` to serve as a global representation of the full text in downstream tasks. However, only a small fraction of typical BERT training is dedicated to sentence-level tasks, such that `[CLS]` representations do not usually perform well at encoding sentence-level semantics (Thakur et al., 2021; Muennighoff et al., 2023). Simply averaging the embeddings across all tokens to obtain a sentence-level representation does not perform well either (Muennighoff et al., 2023). Therefore, more sophisticated pooling (Wang & Kuo, 2020) and post-processing strategies (Li et al., 2020; Su et al., 2021) have been suggested to improve BERT-derived sentence representations. Alternatively, a model pre-trained on the token level can be fine-tuned with a sentence-level objective to improve the sentence-level representations (see below). Usually fine-tuning is performed on the average pooling of all tokens or the `[CLS]` token embedding.

**Decoder-only models**   Decoder-only models are used for text generation and employ masked (as opposed to bidirectional) attention, where each token can only attend to preceding tokens. Different strategies have been proposed to adapt generative models for text-embedding tasks. One approach is to append an instruction before the text and then use either the representation of the last `[EOS]` token (Wang et al., 2024; Li et al., 2025) or the average representation across all tokens (excluding the instruction) (BehnamGhader et al., 2024) as the text embedding. This strategy was originally introduced in encoder-only models (Su et al., 2023), but has increasingly been adopted for generative models. A generative model (with or without additional instructions) can also be fine-tuned with a sentence-level objective (see below). Prior to such fine-tuning, some works remove the causal attention masking, switching the model to bidirectional attention (BehnamGhader et al., 2024; Lee et al., 2025). Additionally, some fine-tuned models use a particular way of pooling such as position-weighted mean pooling (Muennighoff, 2022) or latent attention pooling (Lee et al., 2025).

Note that some tasks that are often used for benchmarking text embeddings can be handled by a generative model without any explicit text embeddings. For example, a classification task can be presented to a generative LLM with a prompt asking to predict the text's label from a given set of labels (Chen et al., 2025). This does not constitute a text embedding.

## 2.2   Contrastive fine-tuning of text embeddings

Whatever the exact model architecture and pre-training are, contrastive learning is commonly used for fine-tuning of text representations. Contrastive learning pulls together pairs of similar texts (*positive pairs*) and pushes apart pairs of non-similar texts (*negative pairs*) in the embedding space (Chen et al., 2020). Contrastive learning approaches differ in how similar texts are defined.

**Supervised contrastive learning**   Here positive pairs are collected based on some explicit notion of similarity. Sentence-BERT (SBERT) (Reimers & Gurevych, 2019) uses a curated dataset of paired texts such as question-answer pairs from Stack Exchange; their most recent model `all-mpnet-base-v2` (2021) was trained on over one billion of such pairs. Similarly, BGE (Xiao et al., 2024), E5 (Wang et al., 2022), and GTE (Li et al., 2023) all undergo contrastive fine-tuning using large datasets of text pairs curated from different sources, such as internet webpages (title-body pairs), scientific papers (title-abstract pairs), or community forums like StackExchange (question-answer pairs). For academic texts, SPECTER (Cohan et al., 2020) and SciNCL (Ostendorff et al., 2022) use citing and cited papers' abstracts to form positive pairs. Sentence-T5 (ST5) (Ni et al., 2022) and Sentence-GPT (SGPT) (Muennighoff, 2022) are derived via contrastive fine-tuning of T5 (Raffel et al., 2020) and GPT (Radford et al., 2018) models on curated datasets of text pairs.

**Self-supervised contrastive learning**   Here positive pairs are generated automatically from unpaired texts, similar to self-supervised learning in computer vision that relies on data augmentations (Chen et al., 2020). SimCSE (Gao et al., 2021) uses two different dropout patterns to form a positive pair of embeddings. This approach has also been used by Liu et al. (2021) and Yan et al. (2021) who additionally investigate other data-augmentation techniques such as randomly masking parts of the input text. The same dropout approach was also adopted by LLM2Vec (BehnamGhader et al., 2024) for fine-tune generative LLMs (after removing attention masking, see above). Alternatively, outputs of two distinct networks can be used to

generate positive pairs (Kim et al., 2021; Carlsson et al., 2021). Further, one can use adjacent chunks of a text as positive pairs; this was applied to train RNN (Logeswaran & Lee, 2018), GPT (Neelakantan et al., 2022), and BERT models (DeCLUTR, Giorgi et al., 2021; Contriever, Izacard et al., 2022a; and CoCondenser, Gao & Callan, 2022). Recently, synthetic generation of positive pairs has been explored leveraging generative LLMs (Zhang et al., 2023; Wang et al., 2024), but this approach comes with the additional computational cost of text generation.

### 2.3 Benchmarks

In benchmarks of sentence transformers, models trained in a supervised way outperform the ones trained only with self-supervision (Thakur et al., 2021; Muennighoff et al., 2023). Among the models of BERT-base size, the BGE-base model (Xiao et al., 2024) with its further modifications and the GTE-base model (Li et al., 2023) are currently among the best performers, overtaking the SBERT's latest `all-mpnet-base-v2` model. Larger models, such as BGE-large or commercial embedding models like `text-embedding-3-large` from OpenAI and `embed-english-v3` from Cohere, outperform BERT-base-sized models, in particular on some of the evaluation tasks.

## 3 Self-supervised contrastive fine-tuning

### 3.1 Augmentations and loss function

We set out to investigate how effectively sentence representations from a language model can be improved through self-supervised fine-tuning alone, and how different augmentation choices influence this improvement. For that, we leveraged a contrastive learning approach analogous to SimCLR (Chen et al., 2020) as our training strategy and compared various augmentation techniques for generating positive pairs, such as text crops (Logeswaran & Lee, 2018; Giorgi et al., 2021; Neelakantan et al., 2022; Izacard et al., 2022a; Gao & Callan, 2022), dropout-based augmentation (Gao et al., 2021), and variations of those (see Section A.1). Importantly, all augmentation approaches in our comparison were constructed in a self-supervised manner, without relying on external notions of similarity or annotated data.

The **cropping augmentation** was set up as follows: for each input text $i$ in a minibatch of size $b$, we cropped out all possible chunks of $t = 2$ consecutive sentences (discarding all sentences under 100 and over 250 characters long) and sampled two chunks, one as the anchor text $a_i$ and one as its positive partner $p_i$. For example, if the abstract of our paper were in the training set, then one positive pair could look like this:

| | | |
|---|---|---|
| Text embeddings, i.e. vector representations of entire texts, play an important role in many NLP applications, such as retrieval-augmented generation, clustering, or visualizing collections of texts for data exploration. Currently, top-performing embedding models are derived from pre-trained language models via supervised contrastive fine-tuning. | $\leftrightarrow$ | This fine-tuning strategy relies on an external notion of similarity and annotated data for generation of positive pairs. Here we study *self-supervised* fine-tuning and systematically compare the two most well-known augmentation strategies used for fine-tuning text embeddings models. |

For the **dropout-based augmentations**, we used the approach of SimCSE (Gao et al., 2021). We split each input text $i$ into groups of consecutive sentences in the same way as for the cropping augmentation, to have similar text lengths for both kinds of augmentations. Then, we sampled one single crop and passed it through the model twice, with two different random dropout patterns applied to it, yielding two different representations that we used as anchor $a_i$ and positive pair $p_i$.

As negative examples for text $i$ we always used the positive partners of all other anchors within the same minibatch $\mathcal{B}$. Unlike some other recent studies, we did not use any *hard negatives* (Cohan et al., 2020; Giorgi et al., 2021; Ostendorff et al., 2022).

During contrastive training, the cosine similarity between the representations of $a_i$ and $p_i$ is maximized, while minimizing the cosine similarities between representations of $a_i$ and $p_j$ for $j \neq i$ within the same

minibatch $\mathcal{B}$. This can be achieved using the InfoNCE loss function (Oord et al., 2018), also known as the normalized temperature-scaled cross-entropy loss (NT-Xent) (Chen et al., 2020). For one sample $i$, the loss is given by:

$$\ell_i = -\log \frac{\exp\left(\mathrm{sim}(\boldsymbol{a}_i, \boldsymbol{p}_i)/\tau\right)}{\sum_{j\in\mathcal{B}} \exp\left(\mathrm{sim}(\boldsymbol{a}_i, \boldsymbol{p}_j)/\tau\right)} \ , \tag{1}$$

where $\mathrm{sim}(\boldsymbol{a}, \boldsymbol{p}) = \boldsymbol{a}^\top \boldsymbol{p}/(\|\boldsymbol{a}\| \cdot \|\boldsymbol{p}\|)$ is the cosine similarity between $\boldsymbol{a}$ and $\boldsymbol{p}$, the vector representations of texts $a$ and $p$. We set the temperature to $\tau = 0.05$ and the batch size to $b = 64$, the largest possible batch size given our GPU memory resources. We trained the network using the Adam optimizer (Kingma & Ba, 2015) with learning rate $\eta = 2 \cdot 10^{-5}$, with linear warm-up and linear decay. See Section A.1 for details on hyperparameter choices.

### 3.2 MTEB performance after SSL fine-tuning

#### 3.2.1 Setup

As the first evaluation suite to assess the sentence representation quality, we used the Massive Text Embedding Benchmark (MTEB) (Muennighoff et al., 2023). It comprises older benchmarks, such as BEIR (Thakur et al., 2021) or STS (Agirre et al., 2012); as well as a wide range of downstream tasks from different modalities, such as clustering, classification, or retrieval.

We fine-tuned a base transformer (i.e. a model primarily pre-trained with a token-level loss) using the SSL training setup outlined in the previous section and compared two augmentation strategies: cropping and dropouts. We chose MPNet (Song et al., 2020; `mpnet-base`) as base model, following SBERT (Reimers & Gurevych, 2019; model `all-mpnet-base-v2` on HuggingFace). This model has `bert-base` architecture with 110 M parameters and uses 768 embedding dimensions. We used mean pooling over all tokens to obtain a single 768-dimensional output vector for each input text, but also compared other pooling strategies (Section A.1, Supplementary Figure S1). Furthermore, we also tested two other base models (BERT and RoBERTa; Devlin et al., 2019; Liu et al., 2019; Supplementary Table S1).

For fine-tuning the base model we used the ICLR dataset (González-Márquez & Kobak, 2024), which consists of 24,347 scientific abstracts of all papers submitted to the ICLR conference within the years 2017–2024. This recently assembled dataset is well-suited for our purposes because it is not part of the MTEB benchmark. Fine-tuning on this dataset therefore enables a fair model comparison and avoids evaluation data leakage, which can potentially lead to inflated performance estimates.

For simplicity, we did not use all MTEB tasks for evaluation, but focused on a subset of English tasks from five different modalities: clustering, reranking, retrieval, STS, and classification. Clustering tasks assess the $K$-means clustering results in the embedding space; retrieval and reranking tasks assess the quality of the nearest neighbors in the embedding space; STS tasks measure how well the embedding represents not only small but also large ground-truth pairwise distances; classification tasks use a logistic regression classifier to assess linear separability of ground-truth classes (see Section A.2 for details). These evaluation modalities quantify different aspects of sentence representation and cover the wide range of MTEB tasks. We used `SentenceTransformers` library that normalizes all text-embedding vectors to have unit norm.

#### 3.2.2 Results

Out of the box, MPNet resulted in poor representations with a block average across modalities of 38.9% (Table 1, column 1). After fine-tuning on the ICLR dataset for a single epoch, the quality of the embeddings markedly improved. Out of the two augmentation strategies, cropping worked much better than dropout: cropping-based fine-tuning outperformed dropout-based fine-tuning in 16/19 tasks (on average 51.8% vs 46.9%, Table 1, columns 3–4), and in the other 3 the difference was small. Cropping-based fine-tuning yielded an improvement in block average score of 12.9 percentage points, but there were large differences between modalities: STS had the largest improvement (21.4 p.p.), followed by retrieval (20.0 p.p.), clustering (10.3 p.p.), reranking (10.1 p.p.), and classification (2.7 p.p.).

Table 1: **MTEB tasks.** Row blocks correspond to clustering, reranking, retrieval, STS, and classification tasks. All values in percent, higher is better. Models in columns 3–4 were fine-tuned on the ICLR dataset.

| | (1) | (2) | (3) | (4) | (5) | (6) | (7) |
|---|---|---|---|---|---|---|---|
| Model | MPNet | SimCSE | MPNet | MPNet | SBERT | BGE-base | BGE-large |
| Augmentations | — | — | Dropout | Crops | — | — | — |
| ArxivClusteringP2P | 27.8 | 35.4 | 33.3 | 38.3 | 48.1 | 48.7 | 48.6 |
| BiorxivClusteringP2P | 23.2 | 30.1 | 31.1 | 32.4 | 39.3 | 39.4 | 39.7 |
| MedrxivClusteringP2P | 22.5 | 28.0 | 29.3 | 30.8 | 35.6 | 33.2 | 32.6 |
| RedditClusteringP2P | 37.4 | 44.7 | 49.5 | 55.9 | 56.6 | 62.7 | 64.7 |
| StackExchangeCl...P2P | 26.3 | 28.8 | 30.2 | 31.3 | 34.3 | 35.2 | 35.0 |
| SciDocsRR | 56.1 | 69.5 | 64.6 | 73.6 | 88.7 | 87.5 | 87.6 |
| MindSmallReranking | 27.5 | 29.3 | 28.4 | 30.2 | 31.0 | 31.2 | 32.1 |
| SCIDOCS | 1.4 | 7.9 | 6.5 | 13.0 | 23.8 | 21.7 | 22.6 |
| ArguAna | 22.2 | 41.4 | 41.9 | 50.6 | 46.5 | 63.8 | 64.5 |
| STS15 | 53.5 | 82.3 | 63.5 | 72.5 | 85.7 | 88.0 | 88.0 |
| STS16 | 50.6 | 77.7 | 66.2 | 76.0 | 80.0 | 85.5 | 86.5 |
| STSBenchmark | 52.0 | 78.6 | 67.9 | 71.7 | 83.4 | 86.4 | 87.5 |
| AmazonPolarityClass. | 66.5 | 72.0 | 63.3 | 62.5 | 67.1 | 93.4 | 92.4 |
| Banking77Class. | 57.4 | 74.4 | 63.3 | 67.7 | 81.7 | 87.0 | 87.8 |
| ImdbClass. | 61.8 | 66.5 | 66.2 | 64.9 | 71.2 | 90.8 | 92.8 |
| MassiveIntentClass. | 55.9 | 60.5 | 53.0 | 53.9 | 69.8 | 72.6 | 74.3 |
| MassiveScenarioClass. | 60.7 | 66.7 | 62.8 | 65.9 | 75.7 | 76.5 | 77.4 |
| MTOPDomainClass. | 75.9 | 82.8 | 82.6 | 85.1 | 91.9 | 93.2 | 94.0 |
| TweetSent...Class. | 52.8 | 52.2 | 51.6 | 49.7 | 55.0 | 59.4 | 59.9 |
| **Block average** | 38.9 | 51.0 | 46.9 | 51.8 | 58.8 | 62.9 | 63.5 |

We repeated the same experiment using two further base models (BERT and RoBERTa) and observed the same results as with MPNet, namely that fine-tuning using cropping augmentation yielded better representations than fine-tuning using dropout (Supplementary Table S1). Representations obtained with cropping-based fine-tuning were better than those obtained via dropout in 16/19 tasks for BERT and in 15/19 for RoBERTa. Compared to the out-of-the-box models, after cropping-based fine-tuning the representations improved by 6.8 percentage points for BERT and by 10.7 p.p. for RoBERTa. The exact ranking of MTEB modalities by improvement size varied between models, but retrieval always showed one of the most pronounced improvements, while in classification tasks there was near-zero change when using BERT and RoBERTa.

The off-the-shelf unsupervised SimCSE model (which is based on dropout fine-tuning) performed similarly to our dropout-based fine-tuned model on all tasks except for the STS and some of the classification tasks, where it was substantially better than all fine-tuned base models (Table 1, columns 2 and 3; Table S1, columns 5 and 8), suggesting that the good performance of SimCSE in those benchmarks may be due to some other fine-tuning choices beyond the dropout augmentation. SimCSE yielded worse results than our cropping-based fine-tuning in the other three modalities, despite having being fine-tuned on two orders of magnitude more data (1 M Wikipedia sentences for SimCSE vs. 24 k samples in our experiments), demonstrating that, at least for most tasks, the choice of augmentation has a greater impact on performance than the amount of training data.

Cropping-based fine-tuned MPNet was 7.0 percentage points below SBERT, which achieved 58.8% performance as block average. Fine-tuned MPNet performed closest to SBERT in retrieval tasks (3.3 p.p. below SBERT) and furthest in STS tasks (9.6 p.p. below SBERT). This demonstrates that cropping-based fine-tuning produced a sentence-level model that showed substantial generalization despite very limited amount

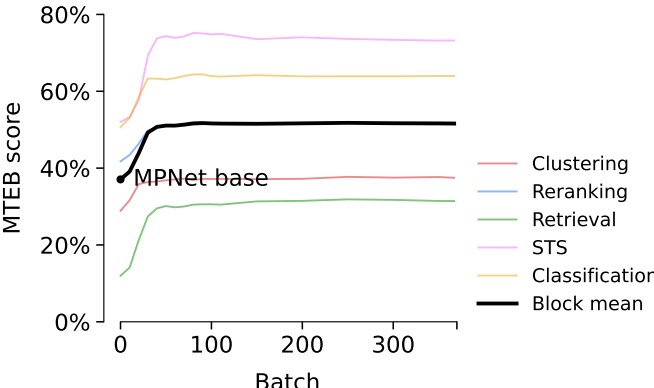

Figure 1: **MTEB score during fine-tuning.** Improvement on block mean MTEB score and individual task modalities within the first fine-tuning epoch on ICLR dataset.

of fine-tuning (∼24 k training samples, ICLR dataset only). Note also that some of the evaluation datasets, e.g. SCIDOCS, arXiv, and StackExchange, formed part of the *training* set of SBERT, possibly biasing SBERT performance estimates upwards.

Additionally, we compared the standard transformer models to two state-of-the-art models: BGE-base (109 M parameters) and BGE-large (335 M parameters) (Xiao et al., 2024) (see Table S2 for the list of all used models). BGE-base is among the best-performing models in public MTEB leaderboard within models of its size. BGE-large is the larger version of BGE-base, and is among the top models overall, on par with `text-embedding-3-large` from OpenAI. In the MTEB tasks we evaluated, both BGE models performed better than SBERT, especially on classification tasks (4.4 p.p. higher in block average) and substantially better than our cropping-based fine-tuned MPNet (11.4 p.p. difference). This suggests that additional training modifications, such as pre-training specifically targeted at sentence embeddings, instruction-based supervised fine-tuning, or hard negative samples, can bring embedding quality further.

### 3.2.3 Cropping-based fine-tuning is very fast

We further analyzed the performance improvement *within* the single fine-tuning epoch. We found that the MTEB block average was improved by around 15 percentage points within the first 100 fine-tuning batches (6 400 positive pairs) (Figure 1). Afterwards, the MTEB score plateaued for all modalities and did not improve any further, and fine-tuning for more than 1 epoch in the same dataset did not bring further improvements. As a caveat, it is possible that one would observe further slower gains in performance with larger amount of training data and longer training duration.

These 100 batches of fine-tuning took only ∼1 min of training time on a single GPU (NVIDIA RTX A6000). In comparison, the top-performing sentence transformer models are typically trained on large datasets, with substantial computational costs and training times. For example, the `all-mpnet-base-v2` SBERT model was trained in a supervised way using over one *billion* text pairs. Even though its performance is higher (58.8%), we could bring the same base model (`mpnet-base`) close to SBERT's performance on some of the tasks in a few minutes of self-supervised training using five orders of magnitude less data.

## 3.3 Representation of a dataset for analysis and visualization

### 3.3.1 Setup

Beyond MTEB, we evaluated the models' ability to generate meaningful representations of a given dataset for data analysis and visualization. This application of text embeddings is essential for understanding the structure of a dataset and identifying outliers or potential data quality issues (González-Márquez et al., 2024;

Table 2: **Representation quality of a given dataset.** $k$NN accuracy of the mean-pooling representation in percent ($k = 10$). We used a 9:1 train/test split for the $k$NN classifier. Columns 1–2, 5–7: off-the-shelf models. Columns 3–4: MPNet fine-tuned on each dataset using cropping and dropout augmentations. Reported values should be interpreted with an error of up to $\pm 1\%$, corresponding to the binomial standard deviation $100\sqrt{p(1-p)/n}$ for test set size $n \approx 2000$ (smallest dataset) and accuracy $p = 0.5$.

| Model | (1) MPNet | (2) SimCSE | (3) MPNet | (4) MPNet | (5) SBERT | (6) BGE-base | (7) BGE-large |
| Augmentations | — | — | Dropout | Crops | — | — | — |
| --- | --- | --- | --- | --- | --- | --- | --- |
| ICLR | 37.4 | 45.7 | 46.8 | 58.9 | 63.3 | 63.3 | 63.5 |
| arXiv | 37.8 | 40.0 | 39.9 | 44.2 | 46.2 | 46.0 | 46.0 |
| bioRxiv | 58.6 | 59.0 | 60.7 | 61.8 | 65.2 | 64.4 | 65.5 |
| medRxiv | 43.5 | 47.2 | 47.8 | 52.4 | 56.8 | 55.7 | 55.3 |
| Reddit | 62.6 | 59.9 | 57.8 | 72.0 | 75.0 | 79.2 | 80.0 |
| StackExchange | 39.3 | 40.7 | 41.6 | 45.6 | 50.6 | 51.4 | 51.5 |
| **Average** | 46.5 | 48.8 | 49.1 | 55.8 | 59.5 | 60.0 | 60.3 |

Anand et al., 2023), yet standard benchmarks often overlook it. Given its practical importance, we included this evaluation to provide a more comprehensive assessment of sentence representation quality across tasks.

We assessed the dataset representation quality through $k$-nearest-neighbor ($k$NN) classification accuracy in the high-dimensional ($d = 768$) embedding space. This metric is computed by classifying each point according to the majority class among its $k$ nearest neighbors, and then comparing predicted class against the true class. We used $k = 10$ with Euclidean distance and non-normalized embeddings, but we obtained similar results using cosine distance (which is equivalent to using normalized embeddings; Table S3). It is a measure of local coherence: it is high if each point's nearest neighbors belong to the same class as the point itself. This metric is particularly relevant for data exploration applications, e.g. visualization or clustering, as many unsupervised learning algorithms for these tasks rely on the $k$NN graph of the data. While this metric depends on the quality of the $k$NN graph similarly to retrieval or reranking tasks from MTEB, it is simpler to evaluate as it requires only class labels rather than annotated samples or ranked neighbors.

To evaluate the self-supervised fine-tuning strategy, we fine-tuned the same base model with the same augmentations and loss as in the previous section, but this time using as training data the *same dataset* that is being represented. That means, for each dataset being evaluated, the base model was fine-tuned separately on that dataset for one single epoch. Note that we purposefully used the entire dataset first for self-supervised training and later for supervised evaluation. As the self-supervised training does not have access to class labels, this setting does not present overfitting issues. Also, the augmentations operate *within* samples (e.g. two fragments of one sample text are encouraged to be similar), and not *across* samples (i.e. two samples from the same class are not encouraged to be similar). We obtained similar results when conducting the same experiment with a train/test split for both self-supervised training and supervised evaluation (Supplementary Table S4).

We performed our fine-tuning experiments on six datasets: the arXiv, bioRxiv, medRxiv, Reddit, and StackExchange datasets from the P2P clustering tasks of the MTEB (Muennighoff et al., 2023), and the ICLR dataset (González-Márquez & Kobak, 2024). The datasets differed in the number of samples (18–733 thousand) and classes (26–610; Table S5). Four of them comprised scientific abstracts from different disciplines, and the other two consisted of internet posts.

### 3.3.2 Results

We found that, on average across datasets, the base MPNet produced representations with low accuracy (46.5%, Table 2) and almost no semantic structure visible in 2D visualizations using $t$-SNE (van der Maaten & Hinton, 2008) (Figure 2). Fine-tuning MPNet for one epoch for each dataset increased the performance

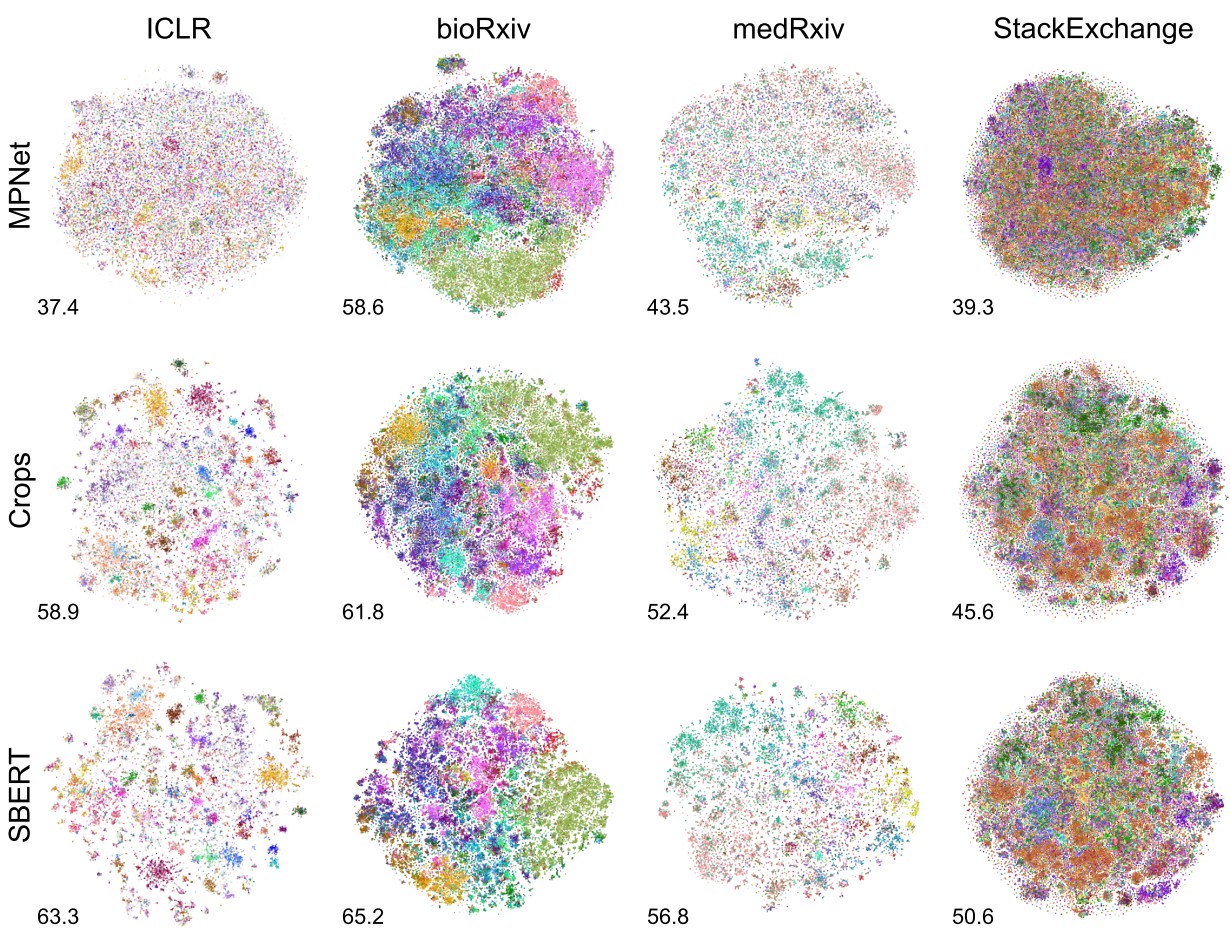

Figure 2: **Dataset visualizations.** $t$-SNE visualizations of MPNet, SBERT, and cropping-based fine-tuned MPNet embeddings of different datasets. Color corresponds to class labels. Numbers show $k$NN accuracy in 768D embedding space. We used openTSNE with default parameters (Poličar et al., 2024). The fine-tuned model was always fine-tuned on the same dataset, as in Table 2.

to 55.8% on average, which was still below SBERT with 59.5%. For some datasets this improvement was particularly large; e.g. the representation of the ICLR dataset improved by over 20 percentage points.

As in Section 3.3, cropping-based fine-tuning outperformed dropout-based fine-tuning in all datasets (on average 55.8% vs 49.1%, Table 2, columns 4–5). The off-the-shelf SimCSE model produced similar representations to our dropout-based fine-tuned model on all datasets.

The difference in performance between the cropping-based fine-tuned MPNet and SBERT (3.7 p.p.) was less prominent in this task than in the retrieval and reranking MTEB tasks, and their 2D visualizations were qualitatively similar (Figure 2). This confirmed that cropping-based fine-tuning produced a sentence-level model that yielded high-quality representations, despite the limited amount of fine-tuning and the lack of supervision.

Furthermore, on three scientific datasets (ICLR, arXiv, medRxiv), the cropping-based fine-tuning matched the performance of SciNCL and SPECTER, two off-the-shelf embedding models specifically designed and trained to represent scientific abstracts (Table S6, columns 8–9), using scientific citations as positive pairs. On the non-scientific datasets (Reddit and StackExchange), cropping-based fine-tuning unsurprisingly outperformed both SciNCL and SPECTER.

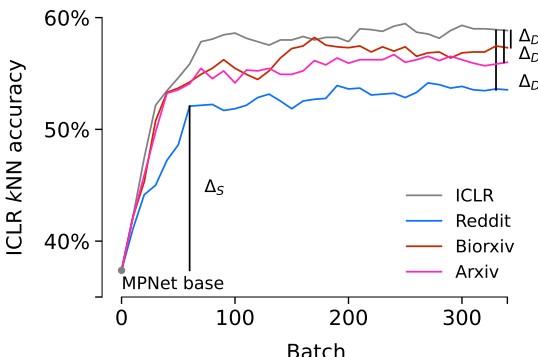

Figure 3: **Sentence vs. domain adaptation.** $k$NN accuracy on the ICLR dataset for MPNet fine-tuned separately on four different datasets (arXiv, bioRxiv, Reddit, ICLR).

Additionally, we evaluated the two state-of-the-art models also used in the previous section: BGE-base and BGE-large. In this task, their performance was closer to SBERT, only minimally surpassing it. The difference in performance between BGE and the cropping-based fine-tuned MPNet was smaller here ($\sim 4$ p.p.), showing that representation quality after self-supervised fine-tuning highly benefits from in-domain training data.

### 3.3.3 Sentence and domain adaptation during fine-tuning

When fine-tuning a base model pre-trained on the token level with a sentence-level contrastive loss on a specific dataset, two mechanisms can contribute to the performance improvement: the model adapting to represent sentences ("sentence adaptation") and the model adapting to the domain of the training data ("domain adaptation"). To disentangle the contributions of these two potential mechanisms, we performed self-supervised fine-tuning on one dataset and evaluated the model's performance on a dataset from a different domain. We used three MTEB clustering P2P datasets from the previous section (arXiv, bioRxiv, and Reddit) as training datasets, and ICLR as the evaluation dataset. For comparable training conditions, we used dataset subsets equal to ICLR's size. We fine-tuned the base model separately on each dataset and measured the $k$NN accuracy on the ICLR dataset. We also fine-tuned and evaluated directly on ICLR for comparison with the setting when both adaptation mechanisms are present.

We found that the ICLR $k$NN accuracy training curve had a similar shape for all training datasets, including the ICLR dataset (Figure 3). Most of the improvement happened within the first 100 batches, and after that the $k$NN accuracy increased only slightly. This agrees with what we observed previously using the MTEB score (Figure 1).

Training on arXiv and bioRxiv yielded better ICLR performance than training on Reddit, likely because scientific abstracts of other disciplines (arXiv and bioRxiv) have greater domain similarity with ICLR abstracts than internet posts (Reddit). This suggests that the performance of sentence embeddings trained with self-supervision decreases for out-of-domain data.

As the domains of Reddit and ICLR datasets are very different, the improvement in ICLR $k$NN accuracy obtained when training on the Reddit dataset must be mostly due to sentence adaptation rather than domain adaptation. This improvement was 14.7 p.p. in our experiment ($\Delta_S$ in Figure 3), which was larger than the difference in final performance between training on the Reddit and on the ICLR datasets ($\Delta_{D_3} = 5.3$ p.p.). Thus, we conjecture that the majority of the improvement in MTEB score when trained on the ICLR dataset (Table 1) was due to generic sentence-level adaptation. This may also explain why the gap between our fine-tuned MPNet and SBERT was larger for MTEB than for the $k$NN accuracy evaluation, since in the first scenario the model was always evaluated on out-of-domain data (compared to the data used for contrastive fine-tuning).

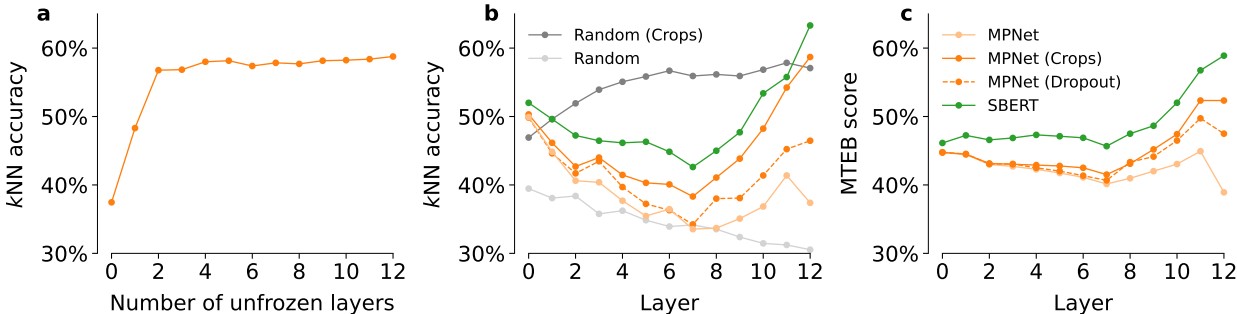

Figure 4: **Representation quality across layers.** **(a)** $k$NN accuracy after fine-tuning MPNet with different number of initial layers frozen. The embedding layer was frozen in all settings. Zero unfrozen layers corresponds to no fine-tuning. **(b)** $k$NN accuracy after each layer for MPNet before and after fine-tuning all layers, for SBERT, for a randomly initialized model with and without training with cropping augmentations. Layer 0 corresponds to the embedding layer. **(c)** MTEB block average score after each layer for MPNet before and after fine-tuning with both kinds of augmentations, and for SBERT. Layer 0 corresponds to the embedding layer. Here the text embeddings were not normalized for the evaluation, unlike in Section 3.2, therefore exact values are slightly different from Table 1.

### 3.4 Self-supervised training without pre-training

To determine whether the token-level pre-training was necessary to achieve good sentence representations of a given dataset, we performed cropping-based contrastive training of the `bert-base` architecture from scratch, without using pre-trained MPNet weights (Table S6, column 10). Here, the performance did not saturate after one epoch, so we continued training for 10 epochs. On average across datasets, the resulting $k$NN accuracy was only 2.6 percentage points below the one we obtained from fine-tuning the pre-trained MPNet, and for some of the datasets there was no noticeable performance difference at all. However, when training the model on ICLR and evaluating on MTEB (setup from Section 3.2 but without classification tasks) the results were much worse compared to using the pre-trained MPNet for fine-tuning (18.9 p.p. difference).

As an additional ablation, we performed the same cropping-based contrastive training of a bare, randomly initialized, embedding layer. This is a direct token embedding model without any transformer architecture whatsoever. Training it for 10 epochs, we obtained dataset embeddings that on average were 3.5 percentage points below the ones obtained from fine-tuning the pre-trained MPNet in $k$NN accuracy (Table S6, column 11). Again, when trained on ICLR and evaluated on MTEB (without classification tasks; non-normalized embeddings), the results were very poor, with a much worse block average compared to the full pre-trained MPNet with additional cropping fine-tuning (13.4 p.p. difference).

These results highlight that even the simple token embedding layer can achieve reasonable embeddings of a given dataset when trained with cropping augmentations (dropout augmentation worked much worse; Table S6). However, the transformer architecture and pre-training knowledge are clearly necessary when generalizing to other tasks and domains.

## 4 Representation quality across layers

To investigate whether fine-tuning the entire MPNet model was necessary to obtain high-quality sentence representations, we performed cropping-based fine-tuning on the ICLR dataset while freezing the embedding layer and various numbers of initial layers. We observed that the performance rapidly improved with the number of unfrozen layers, and fine-tuning only the last 2 out of 12 layers for one epoch was sufficient to reach almost the same value of $k$NN classification accuracy as fine-tuning the full model (Figure 4a). Unfreezing additional layers led only to minor further improvements.

When all layers were unfrozen, the last few layers underwent the largest change during fine-tuning, while the early layers barely changed, in agreement with previous findings in the supervised setting (Merchant et al., 2020; Mosbach et al., 2020). To quantify this, we measured the representation quality after each hidden layer before and after fine-tuning the full model (Figure 4b–c). The gap between them was close to zero for early layers and increased towards the last layers (for both kinds of augmentations: dropout and crops). We observed the same effect when fine-tuning MPNet on other datasets and using the $k$NN evaluation (Figure S2).

Intriguingly, the representation quality across layers in our fine-tuned model as well as in out-of-the-box SBERT formed a U-shaped curve (Figure 4b–c): before fine-tuning the embedding layer representation had the highest $k$NN accuracy and almost the highest MTEB score, and after fine-tuning it was surpassed by only the last layers. Across other datasets, the shape was different and not always U-shaped (Figure S2), but fine-tuned models always exhibited a steep rise in performance towards layer 12. The randomly initialized models did not exhibit this shape: after SSL training, the performance monotonically increased and plateaued half-way through the layers (Figure 4b).

It has been argued that misalignment between the training objective and the downstream task can lead to representation quality peaking in the middle layers (Bordes et al., 2023). This does not explain why the representation quality of non-fine-tuned MPNet sometimes was the highest in the *embedding* layer, but prior work found similar results when evaluating BERT and some other language models on MTEB (Skean et al., 2025). However, none of these models had been fine-tuned for text embeddings, so evaluating their text-embedding performance is not very informative. Our results extend those by Skean et al. (2025) and show that, once fine-tuned for text representation, models' representation quality does increase towards the last layers.

A common practice in computer vision is to have several fully-connected layers (*projection head*) between the last layer and the contrastive loss (Chen et al., 2020), which are discarded after SSL training (*guillotine regularization*), to correct for the misalignment between training and evaluation (Bordes et al., 2023). Our results show that this was not necessary when using the cropping augmentation, as the representation quality peaked in the last layer. However, with dropout augmentation and MTEB evaluation, the representation quality peaked in the *penultimate* layer (Figure 4c). Importantly, it was still lower than with the cropping augmentation. When analyzing MTEB task categories separately, we saw a similar effect with cropping augmentation and some of the MTEB tasks (classification and STS), where representation quality peaked in the penultimate layer (Figure S3).

## 5 Limitations

Our main results focus on two augmentation techniques — cropping and dropout — due to their popularity in the text-embedding literature. We explored additional strategies in pilot experiments (random masking of a fraction of the words, and a combination of that with cropping; Section A.1), but these performed worse compared to cropping and we therefore excluded them from the full analysis. An interesting direction for future work would be to compare cropping augmentation to LLM-generated contrastive pairs based on rephrasing or reformulation of the anchor (Jiang et al., 2022; Wang & Dou, 2023; Abaskohi et al., 2023).

Another direction of future research is to investigate the effect of using hard negatives during training. Hard negatives are negative samples that are semantically or structurally similar to the anchor. They can be identified using external sources of information (such as citations; Cohan et al., 2020; Ostendorff et al., 2022), i.e. in a supervised manner. Recent works have also investigated self-supervised ways of mining hard negatives (Xiong et al., 2021; Liu et al., 2024; Li et al., 2024) and argued that this can lead to improvement performance.

Our comparison of the two augmentation techniques on MTEB tasks relies on models fine-tuned on the ICLR dataset (Section 3.2), representing an out-of-domain evaluation scenario. While we studied performance differences across multiple training datasets in the in-domain setting ($k$NN accuracy, Section 3.3), we did not extend this to the MTEB evaluation. The main reason was that we wanted to avoid any train-test overlap and chose the ICLR dataset for training as it is not part of the MTEB evaluations.

# 6 Discussion

We showed that self-supervised fine-tuning is a powerful strategy for producing high-quality text embeddings with minimal training on in-domain data. To this end, we systematically compared different self-supervised augmentation techniques under the exact same training setup and showed that cropping augmentations were much better than dropout augmentations in all evaluation modalities. This finding is noteworthy because dropout augmentations are one of the most well-cited SSL approaches in the literature on text embeddings (Gao et al., 2021) and continue to be sometimes used for training modern text embedding models (BehnamGhader et al., 2024).

Despite self-supervised fine-tuning substantially improving text embedding quality, there was still a gap compared to the supervised SOTA models. It is unclear whether this gap stems from supervision itself or from the much larger and more diverse datasets on which SOTA supervised models are trained. Such models typically leverage diverse data spanning multiple domains, minimizing out-of-domain scenarios — in contrast to our limited self-supervised training on a single dataset. It remains an open question whether this gap can be bridged with extensive self-supervised training on diverse data or whether supervision is a key element to achieve better performance.

One challenge in model evaluation is the overlap between training and evaluation data across models and benchmarks. Supervised models often construct fine-tuning pairs using the same external notion of similarity (class labels, citation relationships, etc.) that evaluation benchmarks use. For instance, the MTEB Scidoc-sRR task uses citation relationships to determine relevant texts, while SBERT used citation pairs from the same SCIDOCS dataset for fine-tuning, artificially inflating its performance on that task. Although MTEB attempts to address this by removing certain tasks and introducing "zero-shot" scores, they do not give details on which datasets and tasks are problematic for which models. We believe that greater transparency is needed in training and evaluation procedures, to facilitate comparisons between existing models. Our work avoids this problem by comparing different augmentation strategies (crops vs. dropouts) in identical and controlled settings.

### Acknowledgments

The authors would like to thank Carsten Eickhoff for feedback and discussions. This work was funded by the Deutsche Forschungsgemeinschaft (KO6282/2-1) and by the Gemeinnützige Hertie-Stiftung. Philipp Berens and Dmitry Kobak are members of the Germany's Excellence cluster 2064 "Machine Learning — New Perspectives for Science" (EXC 2064, ref 390727645). The authors thank the International Max Planck Research School for Intelligent Systems (IMPRS-IS) for supporting Rita González Márquez.

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

# A    Appendix

## A.1    Augmentation and hyperparameter choices for self-supervised fine-tuning

To select the optimal hyperparameters for our self-supervised fine-tuning, we performed a detailed hyperparameter analysis using the ICLR dataset. We ran the fine-tuning of MPNet for one epoch, and assessed the final $k$NN classification accuracy (Figure S1). In each experiment, all other parameters were kept at their default values described in Section 3.1.

**Pooling**   We compared four different pooling strategies for forming the final sentence representation: average pooling, the classification token `[CLS]`, the separation token `[SEP]` (appended at the end of each input text; similar to `[EOS]` token), and the seventh token (as an example of an arbitrary token). We obtained the best results using the average pooling and `[SEP]` token, with the other two options performing less well (Figure S1a).

When evaluating off-the-shelf models, we always used the mean pooling. On the ICLR dataset, some off-the-shelf models showed slightly higher $k$NN accuracy in the `[SEP]` token representation than using the `[CLS]` token or average pooling, but the difference was small (Table S7). It has recently been shown in a computer vision setting that additional tokens can be used by the transformer model as 'registers' to store high-level features (Darcet et al., 2024). Our results suggest that the same can happen with language models, since the `[SEP]` token often serves as a good sentence representation despite not being explicitly used for training.

**Temperature**   We compared several values of temperature from 0.005 to 5.0, and found that the performance decreased with increasing temperature, with $\tau = 0.005$ and $\tau = 0.05$ yielding similar results (Figure S1a). The value $\tau = 0.5$ used in SimCLR (Chen et al., 2020) performed less well.

**Cropping augmentation**   Our data augmentation consisted of 'cropping out' $t$ consecutive sentences. We used periods ('.') as sentence delimitators, and discarded sentences shorter than 100 characters and longer than 250 characters. We varied the number of consecutive sentences (decreasing the batch size accordingly, to make it fit into the GPU memory) and found that the performance generally decreased with $t$, with the optimal number being $t = 2$ (Figure S1b). Note that in our sampling it was possible for the positive pair of text chunks to overlap (but not to coincide exactly).

**Masking augmentation**   We also experimented with a masking augmentation that replaced a certain fraction of tokens in each input chunk with the BERT's special `[MASK]` token. This was done on top of the cropping augmentation. We found that masking led to deterioration of performance (Figure S1c). Using masking augmentation without cropping (i.e. forming positive pairs by applying two different masking patterns to the entire abstract) did not produce competitive results either.

**Learning rate**   The performance increased with increasing the Adam's learning rate (Figure S1d), until it became too large and the training diverged ($\eta \geq 5 \cdot 10^{-4}$). For the bare embedding layer training, the optimal learning rate was $\eta = 5 \cdot 10^{-1}$.

## A.2    Description of MTEB tasks

Here we provide a brief description of the MTEB tasks that we used for evaluation. We used the MTEB library (`https://github.com/embeddings-benchmark/mteb`) for evaluation. Please see Muennighoff et al. (2023) and references therein for further details.

**Clustering**   Each of the used datasets consists of texts and ground-truth class labels for each text. The texts are embedded using the model, and the embedding vectors are clustered using a mini-batch $K$-means algorithm with batch size $b = 32$ and $K$ equal to the true number of classes. The evaluation score is the so called $V$-measure of agreement between cluster labels and class labels, which is invariant to the permutation of cluster labels. The whole procedure is done separately on several non-overlapping batches and the results are averaged.

**Retrieval** Each of the used datasets consists of a corpus of documents, queries, and a mapping from each query to the relevant documents. The documents and queries are embedded using the model, and the aim is to find the relevant documents within the neighborhood of the query in the embedding space. Neighbors are determined using cosine similarity, and after ranking them, normalized discounted cumulative gain (NDCG) at $k = 10$ nearest neighbors serves as the performance metric. NDCG is obtained by normalizing discounted cumulative gain (DCG), which is defined as:

$$\text{DCG@}K = \sum_{i=1}^{K} \frac{\text{rel}_i}{\log_2(i+1)},$$

where $\text{rel}_i$ is the relevance score of the item at position $i$ (which can be either binary or graded), and $K$ is the number of top results considered. The Ideal DCG (IDCG) is then calculated by sorting the results in the optimal order (most relevant first). Finally, NDCG is obtained by normalizing DCG with IDCG:

$$\text{NDCG@}K = \frac{\text{DCG@}K}{\text{IDCG@}K} \cdot 100\%.$$

**Reranking** Each of the used datasets consists of query texts and a list of relevant and irrelevant reference texts for each query. They are all embedded with the model, and for each query, the text embeddings are ranked based on the cosine similarity to the query embedding. The resulting ranking is compared to the ground-truth ranking, scored for each query via average precision (AP) metric, and averaged across all queries (MAP). Average precision is defined as:

$$\text{AP} = \sum_{k=1}^{n} \frac{P(k) \cdot \text{rel}(k)}{\text{number of relevant documents}},$$

where $P(k)$ is the precision at rank $k$, $\text{rel}(k)$ is the relevance of the item at rank $k$ (in this case only binary; 1 if it is relevant and 0 otherwise), and $n$ is the number of retrieved items. Precision at rank $k$ is defined as:

$$P(k) = \frac{k}{\text{Number of relevant items in top } k \text{ results}},$$

with values going from 0 to 1. MAP values also go from 0 to 1, with higher values being better.

**STS** Each of the used datasets consists of a set of sentence pairs, each pair with a numerical score from 0 to 5 indicating similarity between the two sentences (5 being most similar, and 0 most dissimilar). All sentences are embedded with the model, and for each pair, the embedding similarity is computed using cosine similarity. These embedding similarities are then compared against ground-truth similarities using Spearman correlation.

**Classification** Each of the used datasets consists of a set of text and labels, which can be either topic labels or sentiment labels. The texts from the train and test sets are embedded with the model. The train set embeddings are used to train a logistic regression classifier with 100 maximum iterations, which is scored on the test set. The main metric is linear classification accuracy.

### A.3 Software and Data

The analysis code is available at `https://github.com/berenslab/text-embed-augm`.

# B Supplementary tables and figures

Table S1: **Different base models.** This is an extension of Table 1. Different base models (MPNet, BERT, RoBERTa) before and after fine-tuning with our main augmentations. All values in percent, higher is better. Models in columns 2–3, 5–6, and 8–9 were fine-tuned on the ICLR dataset.

| Model | (1) MPNet | (2) MPNet | (3) MPNet | (4) BERT | (5) BERT | (6) BERT | (7) RoBERTa | (8) RoBERTa | (9) RoBERTa |
| Augmentations | — | Dropout | Crops | — | Dropout | Crops | — | Dropout | Crops |
|---|---|---|---|---|---|---|---|---|---|
| Clustering | 27.4 | 34.7 | 37.7 | 32.3 | 35.1 | 36.4 | 21.9 | 35.4 | 36.8 |
| Reranking | 41.8 | 46.5 | 51.9 | 46.7 | 47.4 | 51.5 | 40.3 | 47.1 | 48.8 |
| Retrieval | 11.8 | 24.2 | 31.8 | 15.6 | 21.3 | 29.4 | 7.2 | 24.0 | 25.0 |
| STS | 52.0 | 65.9 | 73.4 | 57.1 | 59.4 | 69.6 | 59.3 | 69.8 | 70.2 |
| Classification | 61.6 | 63.3 | 64.3 | 65.5 | 62.6 | 64.2 | 63.3 | 63.5 | 64.7 |
| **Block average** | 38.9 | 46.9 | 51.8 | 43.4 | 45.2 | 50.2 | 38.4 | 48.0 | 49.1 |

Table S2: **Details of used models.** Model name, Hugging Face URL, citation, and year.

| Name | Hugging Face | Citation | Year |
|---|---|---|---|
| MPNet | `microsoft/mpnet-base` | (Song et al., 2020) | 2020 |
| BERT | `bert-base-uncased` | (Devlin et al., 2019) | 2018 |
| RoBERTa | `roberta-base` | (Liu et al., 2019) | 2019 |
| SimCSE | `princeton-nlp/unsup-simcse-bert-base-uncased` | (Gao et al., 2021) | 2021 |
| SciNCL | `malteos/scincl` | (Ostendorff et al., 2022) | 2022 |
| SPECTER | `allenai/specter` | (Cohan et al., 2020) | 2020 |
| SBERT | `sentence-transformers/all-mpnet-base-v2` | (Reimers & Gurevych, 2019) | 2021 |
| BGE-base | `BAAI/bge-base-en-v1.5` | (Xiao et al., 2024) | 2024 |
| BGE-large | `BAAI/bge-large-en-v1.5` | (Xiao et al., 2024) | 2024 |

Table S3: **Effect of post-processing transformations.** $k$NN accuracy on the ICLR dataset using different post-processing transformations of the MPNet mean pooling representation, obtained via the Euclidean and the cosine metrics for finding nearest neighbors, before and after fine-tuning the model on the ICLR dataset.

| | Euclidean | Cosine |
|---|---|---|
| *Before fine-tuning* | | |
| Raw | 37.4% | 39.6% |
| Centered | 37.0% | 35.8% |
| Whitened | 5.5% | 18.5% |
| *After fine-tuning* | | |
| Raw | 58.7% | 59.3% |
| Centered | 55.7% | 54.7% |
| Whitened | 36.8% | 55.0% |

Table S4: **Representation of a given dataset; SSL with train/test split.** Score is $k$NN accuracy of the mean pooling representation in percent. Unlike in Table 2, here self-supervised training was only done on the train set; then the classifier was trained on the train set and evaluated on the test set (we used a 9:1 train/test split).

| Model
Pre-trained
Augmentations | MPNet
yes
Dropout | MPNet
yes
Crops |
|---|---|---|
| ICLR | 46.7 | 56.8 |
| arXiv | 38.9 | 44.1 |
| bioRxiv | 60.9 | 61.6 |
| medRxiv | 47.8 | 52.3 |
| Reddit | 59.6 | 71.8 |
| StackExchange | 41.3 | 45.1 |
| **Average** | 49.2 | 55.3 |

Table S5: **Dataset statistics.** Statistics of the datasets used in the experiments of Table 2. The arXiv, bioRxiv, medRxiv, Reddit, and StackExchange datasets are from the P2P clustering tasks of the Massive Text Embedding Benchmark (MTEB) (Muennighoff et al., 2023), and the ICLR dataset is taken from González-Márquez & Kobak (2024). Length refers to the number of characters in each sample text. For the arXiv dataset, we used secondary paper categories (e.g. "cs.AI") as labels.

| Dataset | Samples | Classes | Mean length | Std length |
|---|---|---|---|---|
| ICLR | 24 347 | 46 | 1248 | 316 |
| arXiv | 732 723 | 180 | 1010 | 432 |
| bioRxiv | 53 787 | 26 | 1664 | 542 |
| medRxiv | 17 647 | 51 | 1985 | 843 |
| Reddit | 459 399 | 450 | 728 | 710 |
| StackExchange | 75 000 | 610 | 1091 | 809 |

Table S6: **Representation of a given dataset, additional models.** This is an extension of Table 2. Columns 3–4: MPNet fine-tuned on each dataset using dropout and cropping augmentations, identical to columns 3–4 from Table 2. Columns 8–9: off-the-shelf models. Columns 10–11: Full BERT model and embedding layer (Emb.) trained from scratch for 10 epochs using cropping augmentations.

| Model
Pre-trained
Augmentations | (3)
MPNet
yes
Dropout | (4)
MPNet
yes
Crops | (8)
SPECTER
yes
— | (9)
SciNCL
yes
— | (10)
BERT
no
Crops | (11)
Emb.
no
Crops |
|---|---|---|---|---|---|---|
| ICLR | 46.8 | 58.9 | 56.8 | 57.0 | 57.1 | 57.3 |
| arXiv | 39.9 | 44.2 | 44.2 | 45.2 | 44.3 | 43.4 |
| bioRxiv | 60.7 | 61.8 | 64.8 | 66.4 | 60.6 | 60.7 |
| medRxiv | 47.8 | 52.4 | 52.6 | 52.8 | 44.9 | 49.1 |
| Reddit | 57.8 | 72.0 | 55.2 | 57.3 | 61.8 | 63.6 |
| StackExchange | 41.6 | 45.6 | 41.5 | 42.9 | 45.4 | 45.2 |
| **Average** | 49.1 | 55.8 | 52.5 | 53.6 | 52.4 | 53.2 |

Table S7: **Comparison of pooling strategies.** $k$NN accuracy on the ICLR dataset of different of-the-shelf models using mean pooling, `[CLS]` token, and `[SEP]` token as sentence representations. DeCLUTR and SBERT were originally fine-tuned using mean pooling. SimCSE, SciNCL, and SPECTER were originally fine-tuned using the `[CLS]` token. Best representation is in **bold**, best representation for each model is underlined.

|         | Average | `[CLS]` | `[SEP]` |
|---------|---------|---------|---------|
| MPNet   | 37.4% | 31.8% | 36.3% |
| BERT    | 40.6% | 28.2% | 33.1% |
| SimCSE  | 45.7% | 43.5% | 46.4% |
| DeCLUTR | 50.3% | 45.0% | 34.8% |
| SciNCL  | 57.0% | 56.8% | 57.8% |
| SPECTER | 56.8% | 54.1% | 58.5% |
| SBERT   | **63.3%** | 56.8% | 59.8% |

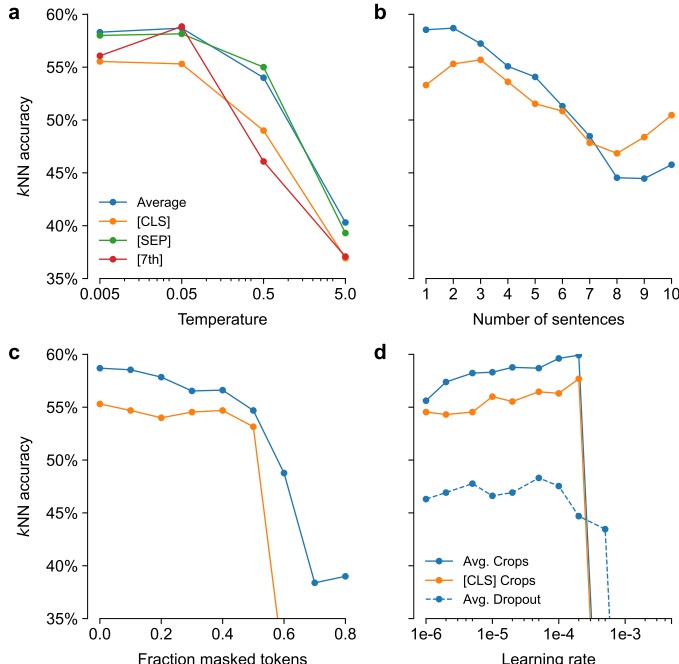

Figure S1: **Hyperparameter tuning.** $k$NN accuracies on the ICLR dataset used for self-supervised training and evaluation, as a function of different hyperparameter values. **(a)** Temperature $\tau$ used to scale the similarities in the loss function. **(b)** Number of consecutive sentences $t$ used in the cropping augmentation. The minibatch size $b$ was adapted depending on $t$ to make it fit into our GPU memory: we used $b = 128$ for $t = 1$; $b = 64$ for $t = 2, 3, 4$; $b = 32$ for $t = 5, 6, 7, 8, 9$; and $b = 16$ for $t = 10$. **(c)** Fraction of masked tokens used in addition of the cropping augmentation. **(d)** Learning rate $\eta$ used by the Adam optimizer.

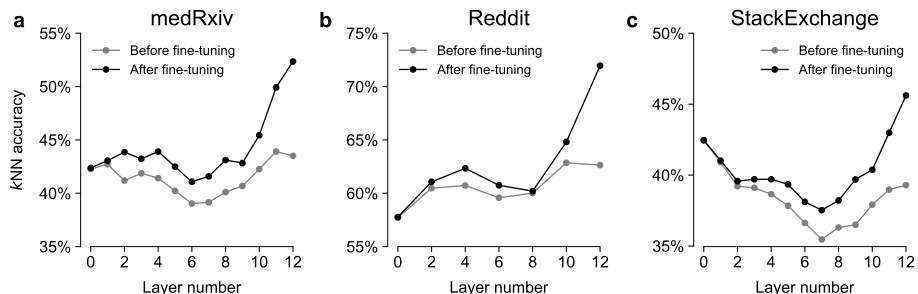

Figure S2: **Representation quality across layers.** $k$NN accuracy after each layer for MPNet before and after fine-tuning in the **(a)** medRxiv, **(b)** Reddit, and **(c)** StackExchange datasets. Evaluation is also done on the respective dataset.

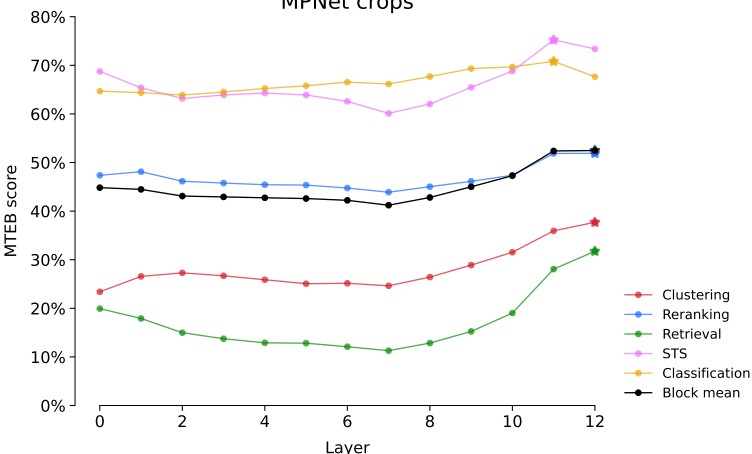

Figure S3: **MTEB score across layers.** MTEB score using the representation after each layer for MPNet after cropping-based fine-tuning (as in Figure 4c), split by task modality. Layer 0 corresponds to the embedding layer. Asterisks highlight the highest performance in each task modality.

