# OpenReview forum: "Cropping outperforms dropout as an augmentation strategy for self-supervised training of text embeddings"
_TMLR — Accepted by TMLR_

### Review · Reviewer_8WK6 · 2025-09-10

**Summary Of Contributions:**

This paper investigates the use of cropping to create positive pairs for document similarity learning, compared to dropout-based construction as well as current larger models.  Two settings are evaluated: (1) training on ICLR abstracts dataset and evaluating on a subset of MTEB tasks; (2) training on a few different arxiv abstract-based tasks and evaluating knn-classification on ICLR.  In both settings, simple cropping constructions with simple models (even a BOW embeddings-only model) provided performance better than dropout methods and sometimes approaching larger models' performance.

**Additional Comments:**

The results for embeddings with no transformer, trained on crops, I think is interesting and could be further explored (Table S5 / sec 3.3.4 "As an additional ablation...") to get more clues as to what is important for these tasks.  In particular:  (1) how much does the vocabulary overlap between tasks, and how is out-of-vocab handled (ICLR data vs bioarxiv, for example)?  (2) Was this juts averaging word embeddings as the comparison model, or was there any small MLP etc on top?  It sounds like it's just averaging embeddings --- in which case it indicates a BOW model is almost as performant on these tasks as most of the more complex methods.  (3)  What are tasks where the BOW model fails, and does the fine-tuning explored in this paper work in these settings?

**Audience:**

Yes

**Audience Explanation:**

I think the presentation of the findings could be adjusted and the investigation further developed, but overall this seem interesting to know about.  At the least it suggests a simple baseline system for text embeddings.

**Claims And Evidence:**

No

**Claims Explanation:**

While the results are suggestive and interesting points of evidence, they fall short of the more general claims that are made.  I would not conclude "using text crops is consistently better than dropout" (claim 1), or "for out-of-domain data ... but for in-domain data ..." (claim 2). These were shown they *can* be the case here, but the evaluation is too limited (only ICLR and small mteb subset) to show more than sometimes this simple method performs well.  Likewise, the claim that "Cropping-based fine-tuning is very fast" (sec 3.2.3) is pretty sweeping, but all that was shown is the initial adjustment from fine-tuning extends well to these small datasets.  Again, it can be a useful method, and starts to make the case that it can be used as a simple and potentially stronger baseline, but it's not clear whether the behavior seen here extends beyond this one dataset.

**Requested Changes:**

See claims section above.  Overall the application experiments are too limited to make broad statements.

---

> ### Author Response · Authors · 2025-11-11
> **Response to Review**
>
> We thank the reviewer for their comments and encouraging assessment.
>
> > Two settings are evaluated: (1) training on ICLR abstracts dataset and evaluating on a subset of MTEB tasks; (2) training on a few different arxiv abstract-based tasks and evaluating knn-classification on ICLR.
>
> Just to clarify: in setting (2), we are training on **six** different datasets (that are **not all abstract-based**, but also include Reddit and StackExchange texts), and evaluate every time on **the same** dataset with a train/test split (and not always on ICLR). This is important to address the criticism below:
>
> > While the results are suggestive and interesting points of evidence, they fall short of the more general claims that are made. I would not conclude "using text crops is consistently better than dropout" (claim 1), or "for out-of-domain data ... but for in-domain data ..." (claim 2). These were shown they can be the case here, but the evaluation is too limited (only ICLR and small mteb subset) to show more than sometimes this simple method performs well.
>
> As explained above, we train and evaluate on a number of different datasets. We did train on other datasets apart from the ICLR dataset, namely all datasets from Table 2, and evaluated kNN accuracy on all of them and observed the same result across all different datasets: cropping-based fine-tuning consistently outperformed dropout-based fine-tuning (on average 55.8% vs 49.1%, Table 2, columns 4–5). The same was true when training on the ICLR data and evaluating ouf-of-domain on different MTEB tasks: cropping-based fine-tuning outperformed dropout-based fine-tuning in all modalities (on average 50.6% vs 45.0%, Table 1, columns 3–4).
>
> Therefore, we believe that our claims are sufficiently backed up by the experiments.
>
> In revision, we added several additional experiments, following suggestions from another reviewer: we added classification tasks from MTEB and also performed fine-tuning of different base models (BERT and RoBERTa in addition to MPNet). We observed the same result: cropping was always better than dropout. We hope that this additional evidence convinces the reviewer.
>
> > Likewise, the claim that "Cropping-based fine-tuning is very fast" (sec 3.2.3) is pretty sweeping, but all that was shown is the initial adjustment from fine-tuning extends well to these small datasets.
>
> We agree that while our experiment in Section 3.2.3 showed plateaued performance after ~50 mini-batches, they do not exclude the possibility of further slower gains after prolonged train on larger datasets. We have now added this caveat to Section 3.2.3.
>
> > (1) [...] how is out-of-vocab handled [when training an embedding-only model from scratch] (ICLR data vs bioarxiv, for example)?"
>
> If we define "out-of-vocabulary tokens" as tokens that have not been seen during training, these tokens would end up with a random embedding. The embedding layer is randomly initialized, which means that all tokens in the vocabulary of the tokenizer are assigned a random embedding. During training, only the embeddings of the tokens that are seen as part of the training data are optimized. Therefore, any "out-of-vocabulary tokens" will be effectively assigned a random embedding.
>
> >(2) Was this juts averaging word embeddings as the comparison model, or was there any small MLP etc on top? It sounds like it's just averaging embeddings --- in which case it indicates a BOW model is almost as performant on these tasks as most of the more complex methods.
>
> Yes, it was simply averaging token embeddings after the initial embedding layer. We agree with the interpretation: on most datasets shown in Table S6, the BOW model was almost as performant as fine-tuned MPNet (and hence close to SBERT etc., see Table 2). One exception here was the Reddit dataset.
>
> >What are tasks where the BOW model fails, and does the fine-tuning explored in this paper work in these settings?
>
> This is a very good question. The BOW embedding layer produced decent representation as measured by the kNN accuracy in the setting where the training and evaluation data were from the same domain. One exception here was the Reddit dataset, where fine-tuning MPNet using crops resulted in 10 percentage point improvement compared to the embedding-only model (and fine-tuning used dropout was even _worse_ than the embedding-only model). Furthermore, on the MTEB tasks, where training data (ICLR) and evaluation data (MTEB datasets) were very different, the embeddings produced by a BOW model unsurprisingly performed very poorly, with the block average much worse compared to using the full pre-trained and fine-tuned MPNet (13.4 p.p. difference).
>
> Our updated manuscript includes tracked changes.

---

### Review · Reviewer_Jwws · 2025-09-13

**Summary Of Contributions:**

This paper shows that cropping augmentation outperforms dropout (SimCSE) for self-supervised text embeddings. Cropping yields stronger results on MTEB and in-domain datasets, improving quality rapidly with minimal data and compute. Most gains arise from sentence-level adaptation, concentrated in the final transformer layers, allowing efficient partial fine-tuning. While cropping-based models approach supervised baselines like SBERT and SciNCL in-domain, they remain weaker on out-of-domain data. Overall, cropping is a simple and efficient augmentation that produces high-quality embeddings and challenges the dominance of dropout-based self-supervised methods.

Strengths:
1. The paper is well-structured and easy to follow.
2. The problem investigated is interesting and important for training language models.
3. The experimental results reveal some interesting findings, including (1) cropping is better than dropping, (2) the layer-wise adaptation is good for in-domain adaptation but not out-of-domain adaptation.

Weaknesses:
1. Despite cropping’s strong in-domain results, the method still underperforms supervised models like SBERT and BGE on out-of-domain tasks.
2. The study focuses mainly on cropping vs. dropout, leaving other promising augmentations (e.g., paraphrasing with LLMs, semantic reformulations) underexplored.
3. Some benchmarks overlap with data used in supervised models (e.g., SBERT with SCIDOCS), making absolute performance comparisons less conclusive and highlighting the need for more transparent evaluation settings.

**Audience:**

Yes

**Audience Explanation:**

Pre-training language models is an important topic in the machine learning community. The paper investigates the tricks in training language models.

**Broader Impact Concerns:**

This paper shows that simple cropping augmentations can make self-supervised text embeddings more efficient and competitive, potentially reducing reliance on costly supervised datasets for building high-quality language representations

**Claims And Evidence:**

Yes

**Claims Explanation:**

The paper conducts a comprehensive evaluation to support the claim made in the paper.

**Requested Changes:**

See weaknesses.

---

> ### Author Response · Authors · 2025-11-11
> **Response to Review**
>
> We thank the reviewer for their comments. We are happy that the reviewer thought our paper "is well-structured and easy to follow'', "[t]he problem investigated is interesting and important'', and ''[t]he experimental results reveal some interesting findings''.
>
> >Weaknesses:
> >1. Despite cropping’s strong in-domain results, the method still underperforms supervised models like SBERT and BGE on out-of-domain tasks.
> >2. The study focuses mainly on cropping vs. dropout, leaving other promising augmentations (e.g., paraphrasing with LLMs, semantic reformulations) underexplored.
> >3. Some benchmarks overlap with data used in supervised models (e.g., SBERT with SCIDOCS), making absolute performance comparisons less conclusive and highlighting the need for more transparent evaluation settings.
>
> We fully agree that these are important points! In fact, these three aspects are exactly the ones that we discussed as limitations of our work in our discussion section. Point 1 is discussed in the penultimate paragraph, point 2 in the second paragraph, and point 3 in the last paragraph.
>
> In our opinion, point 3 is not so much a limitation of our work, but rather a generic challenge for the field of text embeddings. Points 1 and 2 are important directions for future work.
>
> Following comments by another reviewer, we have now added one more future research direction (concerning hard negatives) to the Discussion section. Please see our Discussion section for more details on all these points.
>
> Our updated manuscript includes tracked changes.

---

### Review · Reviewer_db5g · 2025-10-24

**Summary Of Contributions:**

This paper compares two augmentation strategies for positive pair generation in
contrastive learning of text embedding — dropout and text cropping. Authors show that cropping consistently outperforms dropout on some MTEB tasks and several in-domain datasets, achieving results that are only
marginally below supervised SOTA. The pape ralso provides analysis on semantic structure of resulting embeddings, assessing dataset representation quality using KNN, and analysis on what causes the improvements in quality of text embeddings after fine-tuning (is it "sentence adaptation" or "domain adaptation"). Further, authors investigate which layers produce embeddings of the best quality, showing that fine-tuning only the last layers of a model lead to similar fine-tuning performance.
Overall, the paper provides strong and consistent evidence that cropping-based fine-tuning outperforms dropout-based one across modalities in the setup of the paper, while being fast. This can be useful for practitioners who build self-supervised pipelines

Strenghts:
- Strong evidence that cropping outperforms dropout as an augmentation strategy for self-supervised fine-tuning of text embeddings in the given setup. That might be useful for practitioners who build self-supervised pipelines
- The overall setup is clearly defined and seems to be reproducible (maybe except cropping algorithm parameters, e.g. overlap rate control, sentence boundary detection strategy, etc.). Release of the code would be beneficial (there is a phrase "The analysis code is available at URL" in the paper, but URL is invalid). Whole setup is controlled, effects of augmentations are isolated, which makes results trustable
- Good analyses: 1) analysis of semantic structure of resulting embeddings 2) analysis on causes of improvements in quality of text embeddings after fine-tuning (is it "sentence adaptation" or "domain adaptation"), with evidence that SSL might underperform in out-of-domain settings 3)analysis on which layers produce embeddings of the best quality 4) evidence that fine-tuning only the last layers of a model lead to similar fine-tuning performance 5) analysis of speed (cropping-based fine-tuning is fast)


Weaknesses:
- Only one model (MPNet), few MTEB subtasks (clustering, reranking, retrieval, and STS), and two augmentation strategies (cropping and dropout) are considered. There is at least classification task that is different from mentioned in the paper, and, in my opinion, is worth testing on. Or a discussion on why these four subtasks are good indicators of overall performance would be beneficial. Regarding augmentation strategies, authors mentioned that they did not use hard negatives. I wonder if hard negatives could boost performance and influence results, as from my experience with contrastive learning in images this might be the case.
- Abstract is hard to read: it begins with discussion of text embeddings, then jumps to images: "This contrasts with computer vision, where self-supervised trainingbased on data augmentations has demonstrated remarkable success.", and it is hard to understand if the next sentence is about text or visual domain. I would also think that SSL strategies for text embeddings are worth exploring regardless of their success in vision domain, as strategies used in these domains are not directly applicable  between domains, and, at least for me, this passage about images was confusing

**Audience:**

Yes

**Audience Explanation:**

I think that the topic discussed in the paper (comparison of augmentation strategies for SSL for text embeddings) and results can be useful to practitioners doing SSL for text embeddings

**Claims And Evidence:**

Yes

**Claims Explanation:**

Paper is clearly written, experimental setup and analyses are also clear

**Requested Changes:**

My biggest concerns are that only one model is used (MPNet), and augmentation setup is highly limited (for example, usage of hard negatives is not discussed, though might have effect on performance). In my opinion, for the results of the paper to be clearly convincing to practitioners doing SSL, results should be obtained on multiple models and all the major parameters of augmentations (which is, in this case, usage of hard negatives ). However, I am not quite familiar with what models are typically used for SSL on text embeddings, and what are typical parameters of augmentations in this area, so I would be happy to discuss these points with the authors

---

> ### Author Response · Authors · 2025-11-11
> **Response to Review (I)**
>
> We thank the reviewer for their positive comments and very helpful suggestions!
>
> > My biggest concerns are that only one model is used (MPNet) [...]
>
> This is a good point. As suggested by the reviewer, we now tested two further base models: BERT and RoBERTa. We used the setting of our Table 1, where we fine-tune on ICLR abstracts and evaluate on MTEB. The results match what we saw with MPNet, namely that fine-tuning using crops works better than fine-tuning using dropout.
>
> Representations obtained with cropping-based fine-tuning were better than those obtained via dropout in 16/19 tasks for BERT and in 15/19 for RoBERTa. Compared to the out-of-the-box models, after cropping-based fine-tuning the representations improved by 7.0 percentage points for BERT and by 10.4 p.p. for RoBERTa. The exact ranking of MTEB modalities by improvement size varied between models, but retrieval always showed one of the most pronounced improvements, while in classification tasks there was near-zero change when using BERT and RoBERTa.
>
> We added this to the Results section and as Supplementary Table S1.
>
> | Model              | MPNet   | MPNet   | MPNet | BERT | BERT    | BERT  | RoBERTa | RoBERTa | RoBERTa |
> |--------------------|---------|---------|-------|------|---------|-------|---------|---------|-------|
> | Augmentations      | ---     | Dropout | Crops | ---  | Dropout | Crops | ---     | Dropout | Crops |
> | Clustering         | 27.4    | 34.7    | 37.7  | 32.3 | 35.1    | 36.4  | 21.9    | 35.4    | 36.8  |
> | Reranking          | 41.8    | 46.5    | 51.9  | 46.7 | 47.4    | 51.5  | 40.3    | 47.1    | 48.8  |
> | Retrieval          | 11.8    | 24.2    | 31.8  | 15.6 | 21.3    | 29.4  | 7.2     | 24.0    | 25.0  |
> | STS                | 52.0    | 65.9    | 73.4  | 57.1 | 59.4    | 69.6  | 59.3    | 69.8    | 70.2  |
> | Classification     | 52.4    | 53.9    | 58.0  | 57.1 | 52.3    | 56.8  | 56.7    | 54.2    | 56.5  |
> | **Block average**  | **37.1** | **45.0** | **50.6** | **41.8** | **43.1** | **48.7** | **37.1** | **46.1** | **47.5** |
>
> > [...] few MTEB subtasks (clustering, reranking, retrieval, and STS) [...] There is at least classification task that is different from mentioned in the paper, and, in my opinion, is worth testing on.
>
> This is a reasonable suggestion. Originally we did not want to include linear classification tasks because they train a linear classifier on top of embeddings, but we have now decided to add it anyway. We added some of the MTEB classification tasks to Table 1 and again saw that crops were better than dropout on average (58.0 vs. 53.9).
>
> |                                        |   MPNet |   Dropout |   Crops |   SBERT |
> |:---------------------------------------|--------:|----------:|--------:|--------:|
> | AmazonPolarityClassification           |    66.5 |      63.3 |    62.5 |    67.1 |
> | Banking77Classification                |    57.4 |      63.3 |    67.7 |    81.7 |
> | ImdbClassification                     |    61.8 |      66.2 |    64.9 |    71.2 |
> | MassiveIntentClassification            |    26.1 |      24.1 |    35.4 |    42.8 |
> | MassiveScenarioClassification          |    26.1 |      26.5 |    40.7 |    52.1 |
> | MTOPDomainClassification               |    75.9 |      82.6 |    85.1 |    91.9 |
> | TweetSentimentExtractionClassification |    52.8 |      51.6 |    49.7 |    55.0 |
> | **Average**                            |    **52.4** |      **53.9** |    **58.0** |    **66.0** |
>
> We noticed that in 3 out of 7 classification tasks (AmazonPolarityClassification, ImdbClassification, and TweetSentimentExtractionClassification) crops were slightly worse than dropout, and in two of them, both crops and dropout were worse than pre-trained MPNet. These three tasks are sentiment classification tasks (positive or negative) of e.g. Amazon reviews and tweets. Our intuition is that the domain of these tasks is so different from the domain of our training data (ICLR abstracts), that finetuning the models on it hurts downstream performance. Apart from that, it is known in the literature that while a text embedder can perform well in capturing the general semantics of a text, it is a harder task to capture the fine details of sentiment or principles (Devlin et al., 2019; Liu et al., 2022), so our training may not be optimaly tailored for this specific downstream task.
>
> In the other classification tasks, where the text has a class label that relates to its topical content, the results matched those of the other modalities: crops are significantly better than dropout.
>
> We added these classification tasks to Table 1 in the paper and incorporated them into Figure 1.
>
> > only [...] two augmentation strategies (cropping and dropout)
>
> Note that we did look into some additional augmentation strategies in Section A.1. Namely, we used random masking of a fraction of the words, and a combination of that with cropping. They performed worse than cropping.

---

> > ### Author Response · Authors · 2025-11-11
> > **Response to Review (II)**
> >
> > > Regarding augmentation strategies, authors mentioned that they did not use hard negatives. I wonder if hard negatives could boost performance and influence results, as from my experience with contrastive learning in images this might be the case.
> >
> > We agree that hard negatives may boost performance, but if so, this would likely affect the performance of any augmentation strategy. Our main goal was to compare dropout augmentation with cropping augmentation, and so for simplicity we did not use hard negatives (hard negative mining in a self-supervised manner comes along with an increased computational complexity). While it may be interesting to explore how incorporating hard negatives may improve performance, we do not expect the augmentation ranking (dropout vs crops) to change.
> >
> > We have now added a paragraph in our discussion about this, citing some relevant papers using hard negatives.
> >
> > >  Abstract is hard to read: it begins with discussion of text embeddings, then jumps to images [...] and, at least for me, this passage about images was confusing.
> >
> > Thanks for pointing this out. We rephrased the abstract accordingly and hope that the content of the paper comes across more clearly now.
> >
> > > The overall setup is clearly defined and seems to be reproducible (maybe except cropping algorithm parameters, e.g. overlap rate control, sentence boundary detection strategy, etc.).
> >
> > We had details about the cropping augmentation in the Appendix section A.1, but we now further extended it to increase clarity.
> >
> > > there is a phrase "The analysis code is available at URL" in the paper, but URL is invalid
> >
> > We will certainly make the code available upon deanonymization as a Github repository. Please let us know if you want to look at the code during review, and then we will prepare an anonymized version of it.
> >
> > Our updated manuscript includes tracked changes.

---

### Decision · Action_Editor_5waU · 2026-01-22

**Recommendation:** Accept with minor revision

**Additional Comments:**

Please add an explicit limitation section explaining the caveat of having a single training set.

Also, justify why you consider only these two augmentation methods.

**Audience:**

Yes

**Audience Explanation:**

This paper will be interesting and useful to a large audience working with encoder and embedding models.

**Claims And Evidence:**

Yes

**Claims Explanation:**

This paper explores two different techniques for data augmentation in semi-supervised learning of text embeddings: cropping and dropout. Authors show that cropping consistently outperforms dropout. While all the reviewers find this paper to be useful, there are concerns about single training set and only 2 different augmentation techniques.

---

> ### Author Response · Authors · 2026-02-27
> **Changes in the camera-ready version**
>
> Dear Action Editor,
>
> We have incorporated all the requested revisions and included some additional analyses to further support our results.
> Specifically, we:
> - added a dedicated Limitations section addressing the mentioned caveats (comparison of only two augmentations, use of a single training set for the MTEB evaluation, and the absence of hard negatives);
> - expanded the Related Work section to reflect recent developments in the field;
> - extended Figure 4 by adding panel (c) (layer-wise MTEB evaluation), and revised Section 4 accordingly to discuss these new results.
>
> We hope that these revisions satisfactorily address the requested changes, and we sincerely appreciate your and the reviewers feedback throughout the review process.